# Review of Frameworks for Assessing the Strength of the Sanitation Economy and Investment Readiness

**DOI:** 10.3390/ijerph22121868

**Published:** 2025-12-15

**Authors:** Guy Hutton, Sue Coates

**Affiliations:** 1Innate Values Ltd., 7 Woodlands Park, Bath BA1 7BQ, UK; 2Sanitation & Hygiene Fund, 15 Chemin Louis-Dunant, 1202 Geneva, Switzerland

**Keywords:** sanitation, hygiene, enabling environment, sanitation economy, investment readiness, market attractiveness, framework, assessment, monitoring

## Abstract

An improved understanding of the sanitation enabling environment and status of market development (“sanitation economy”) is crucial not only for advancing national and global sanitation goals, but also for attracting the financing necessary to drive meaningful progress in low- and middle-income countries (LMICs). This need is particularly pressing as the sanitation sector faces a significant funding gap that must be bridged to meet the growing demands for sanitation services, infrastructure, and innovation. This paper reviews frameworks that assess the sanitation economy in LMICs with the aim of informing the development of more impactful future frameworks and the wider application of existing frameworks. Frameworks were identified through internet search and interviews with representatives of international sanitation sector organisations and universities. Thirty-nine frameworks were identified that have been or are currently being used in sanitation. Frameworks are diverse in the structure they adopt, their focus areas, the number of indicators, the number of countries covered, the frequency with which they have been applied, their reliance on primary versus secondary data sources, and their uptake and impact. Overall, use of the frameworks has been piecemeal and sporadic in LMICs. Only few frameworks have been picked up and applied by another organisation, although the results of some frameworks are widely used and cited. To ensure future efforts to measure and monitor the sanitation economy are evidence-based and make the best use of limited resources, frameworks currently in use should be independently evaluated and there should be greater collaboration and adoption of common frameworks.

## 1. Introduction

The potential market for sanitation goods and services in low- and middle-income countries (LMICs) is very sizable: every household and institution needs sanitation services, and many are ready to pay for better sanitation services. Globally, 1.5 billion people do not use a basic sanitation facility at home, half of whom live in sub-Saharan Africa [1]. An even larger number—3.4 billion people—do not use a safely managed sanitation facility at home, over a billion of whom live in Central and South Asia [1]. In addition, sanitation is forecast to have significant market value over the longer-term: those with sanitation facilities need to operate, maintain and renew hardware; many households and institutions regularly purchase toilet paper, soap and detergent; and the value of nutrients, energy and water recovered from faecal sludge and wastewater [2,3].

Under the vision of the Sustainable Development Goals (SDGs), the ‘sanitation economy’ encompasses different parts of the sanitation value chain to achieve the ‘safely managed sanitation’ standard monitored under SDG Target 6.2—from the toilet and containing excreta, to emptying, transport, and treatment, followed by either disposal or reuse [4]. Although not monitored under the SDGs, the sanitation economy includes consumables such as toilet paper, hand sanitizer and detergent. In line with the SDGs, many sanitation organisations now emphasise the importance of connecting the biocycle, which involves combining multiple forms of biological waste, recovering nutrients and water, and creating value-adding products [5]. The ‘circular’ sanitation economy includes products such as biogas, electricity, compost and reclaimed water [6]. The ‘smart’ sanitation economy will increasingly include sensor and smart technologies, data analytics and mobile applications [6]. The sanitation economy also includes toilet installation, renovation, cleaning and maintenance services [6].

While the size of the overall sanitation economy is potentially very significant, current investments in sanitation are massively insufficient in most countries to meet national targets. The World Health Organization (2022) reported that less than one in six countries had sufficient funding from all sources to reach national sanitation targets, out of 100 LMICs that reported on this indicator [7]. The cause of the difference between the potential size of the sanitation economy (or market) and the actual budget allocations and customer spending has been the subject of considerable debate. Some major, interconnected causes of the low rates of investment include: (a) public funds from government and donor sources are severely constrained with respect to the investment need [7]; (b) tariffs for emptying and treatment services do not cover the costs of providing the service, which disincentivises private sector providers from expanding their businesses [7,8]; (c) willingness to pay for sanitation products and services does not cover a major share of unserved customers [9]; and (d) private investments (e.g., from banks or individual investors) in sanitation appear risky [10].

Today’s Official Development Assistance (ODA) environment continues to shift from traditional aid models towards more sustainable, market-driven solutions to global challenges, which include blended finance, private sector involvement, and result-based funding. Therefore, attention has increasingly turned to the potential of the private sector to expand investments in sanitation goods and services, within a public regulatory framework (e.g., [11]). Private investment here refers to either an outside investor providing debt or equity capital to a sanitation business, or a sanitation business using its own funds or cross-subsidies to finance an expansion or service improvement.

Several constraining factors combine to make such private investments appear highly risky [10,12,13]. For example, sanitation services such as treatment plants or emptying services require large upfront capital expenditures with uncertain revenue streams to repay loans or reward equity investments. Many of these investments have proven to be unsustainable given the lack of micro-, small- and medium-sized enterprise (MSME) eco-system to support the value chain. Loan finance is difficult to access due to lack of creditworthiness of sanitation enterprises and high interest rates in most LMICs. Furthermore, there remain legal or regulatory limitations or uncertainties in the use of private capital in some countries [14]. The multiple challenges faced in mobilising investments for sanitation therefore requires a comprehensive analysis of the sector to better understand how the right foundations can be built and what investment sources are most appropriate in a given context, and when [15].

While these barriers to private finance constitute a major reason for the lack of private investment in sanitation in LMICs, another connected reason is that potential investors do not fully understand the sanitation economies and the broader systems within which they operate. Furthermore, investors may not be connected to the sanitation sector to see progress in the investment environment [16]. In other words, there is an information asymmetry which contributes to the market failure.

One way to facilitate the sanitation economy and stimulate investment is to make available better information to stakeholders that play a role in stimulating or making investments. For example, the Continental Africa Water Investment Programme (AIP) High-Level Panel Report states that inefficiencies need to be identified and addressed to increase water and sanitation investments, which requires improved information and diagnostics, including, critically, better reflecting the perspective of the private investor [17]. The Economist Magazine states that de-risking investments, increasing returns and unlocking sanitation markets are critical to attract private-sector finance, as highlighted at an Africa Finance Ministers’ meeting hosted by the Sanitation and Water for All (SWA) partnership [18]. Therefore, having more solid evidence on the status of the sanitation economy will allow investors to know in which countries or sub-national contexts sanitation is more ready for investment and to clarify the best way of blending public and private investments to have the greatest, most equitable impact on the unserved population.

The sanitation ‘economy’ is more than just a question of understanding the market size for different sanitation sub-markets or parts of the sanitation value chain. In this paper, sanitation ‘economy’ also encompasses the enabling factors and the bottlenecks that exist in the sanitation market (i.e., the supply and demand for sanitation goods and services) as well as in the broader policy environment. In other words, it helps understand why the sanitation market is or is not growing, in terms of both coverage indicators and expenditures.

The status of the sanitation economy can be expressed in terms of its ‘maturity’, a term which has a strong foundation in product market development. In the field of sanitation, the concept of economy or market ‘maturity’ is not well defined, and so a definition is informed by related fields [19]. Maturity in this context describes “the extent to which the sanitation economy has been developed and is able to deliver quality sanitation services to the entire population sustainably and at an affordable cost”. Levels of maturity then need to be defined.

To date, many conceptual frameworks and associated tools have attempted to measure different components of the sanitation economy, typically with the objective of identifying how it can be strengthened. The literature is rich and extensive, and an important part of it is reflected in the frameworks and tools presented in this paper. References to the enabling environment for water, sanitation and hygiene can be traced back to the early 2000s [20] and later adopted by international organisations such as the World Bank [21] and through global monitoring initiated by WHO in 2008 [22]. A framework is defined as a structure that characterises the sanitation economy and which may include indicators, while a tool is typically based on a framework and includes indicators, provides data collection and indicator scoring methodologies, and it includes a user interface to support the application of the framework.

Essentially, analytic frameworks have focussed on measuring different components of the sanitation economy such as the status of the policy enabling environment, market development, capacities, and financing. However, as will be detailed in this paper, different frameworks have categorised these in a multitude of ways, with variation in the depth and breadth of aspects of the enabling environment included. Amjad et al. (2015) define an enabling environment for drinking water services to include “a favourable culture of internal coordination and communication; policy and institutional behaviour that guides behaviour of water service providers with clear and enforceable service standards, and resources to provide effective water services” (page 1503) [23]. Specific for sanitation, Parkinson et al. (2021) include government support, legal framework, institutional arrangements, skills and capacities, financial arrangements and socio-cultural acceptance as key components of the enabling environment [24]. More recently, ‘systems thinking’ in WASH has received considerable attention with a focus on assessing the dynamics of the interconnected factors that affect WASH outcomes [25]. How others have characterised the sanitation enabling environment is covered in Section 3.

Given the different ways in which the sanitation economy has been defined and assessed, the purpose of this paper is to compare previous frameworks and tools in how they have been structured, identify gaps in aspects of the sanitation economy included, and assess how they have been applied and with what results. The general aim of the paper is to inform the development of more impactful future frameworks as well as the wider application of existing frameworks. The specific aim of the paper is—in a future step—to inform the design of a new framework which draws on the lessons from existing frameworks to enable a comprehensive assessment of the sanitation economy, leading to the identification of targeted actions for specific stakeholders to strengthen the sanitation economy.

The paper answers four main research questions (with sub-questions):What frameworks have been used to better understand and measure the maturity and investment readiness of the sanitation economy? (How have frameworks evolved since they first appeared in the early 2000s?)How are frameworks structured and which indicators have been selected to assess the sanitation economy? (What are the commonalities and differences between frameworks?)What data have been collected by these frameworks? (In which countries and sub-national contexts have they been applied? What is the frequency of data collection? How robust are the data?)How has evidence from the frameworks been presented, published, disseminated and used? (Have the results been accepted by sanitation authorities? Has impact been measured?)

## 2. Materials and Methods

Frameworks were identified through pre-existing knowledge of their use in the sanitation sector, through interviews with representatives of over 50 international WASH sector organisations (see Acknowledgements), and through PubMed, Google and Google Scholar searches using several key terms in combination with ‘sanitation’: ‘enabling environment’, ‘economy’, ‘governance’, ‘market strengthening’, ‘market development’ and ‘market assessment’. Given the very large number of results, the first thousand items in each search engine were examined (conducted in January 2025). The scope of the review is any framework (which may or may not include an associated tool) that focuses exclusively on sanitation or that includes sanitation as one among other key pillars. Included frameworks are those that present novel characterisations of the sanitation market or policy enabling environment, including sanitation finance, that enables assessment of the status, performance, strengths and weaknesses of the sanitation economy. Given this scope, methodologies that focus on estimating sanitation service coverage, estimating costs or benefits of meeting sanitation targets, sizing the sanitation market, planning sanitation interventions or programmes, proposing service delivery models, conducting climate risk assessment, or assessing water or wastewater utility performance are excluded. Frameworks that consist of a combination of other frameworks are excluded, while the frameworks they contain are assessed for relevance.

Once the frameworks were identified and inclusion/exclusion criteria applied, documents relating to frameworks meeting the inclusion criteria were downloaded from the internet. These included documents describing the framework, methodologies for using the framework, results from application of the framework in one or more countries, and assessments or evaluations of how the framework has been used. Initially, information was extracted from available documents to populate a data sheet for each framework that included: the framework structure, its indicators, data generated, data collection methodologies, presentation of results, use of results in defining responses or influencing decisions, lessons from the framework roll out, and reference materials. Next, organisational focal points for each framework were informed about the aims of the review and intention to publish results and invited to provide additional information to complete the data sheet. Feedback was received on the frameworks as a written response whereby the data sheets were edited in tracked changes, as well as the sharing of additional documentation from which information was extracted. The revised data sheets were reshared with organisational focal points to provide opportunity for final adjustments and informed again of the intention to publish it. No objections were made to publishing the edited data sheets shared.

Data were analysed by compiling and comparing key features of each framework (structure, number of indicators), utilisation (how many applications in how many countries, and latest application), documentation of framework applications, impact and lessons learned, which are described in this paper. The review concludes with key lessons for how these or new frameworks can be better defined and become more impactful with respect to understanding and strengthening the sanitation economy.

## 3. Results

### 3.1. Overview of Findings

Thirty-nine frameworks were identified that have been or are currently being used for assessment of the sanitation economy (see Table 1). Prior to the internet searches, thirty-four frameworks were already known about through previous work of the authors, while three additional frameworks were identified by the paper’s peer reviewers and other informants, and two additional frameworks identified through internet search. Seventeen—almost half—of the frameworks are focused exclusively on sanitation, eighteen focus on water, sanitation and hygiene (WASH), and four include both water resources management and WASH.

Note that due to the proliferation of City-Wide Inclusive Sanitation (CWIS) initiatives across many sector organisations, four frameworks are presented for illustrative purposes [26]. Schertenleib et al. (2021) provide a historical account of the development of urban sanitation frameworks and a selective overview of seven approaches and six tools [27]—hence this account is not repeated here. Also, many publications explore systems approaches to WASH [25,28] and sustainability of WASH services [29], and these frameworks were only included where they provide an original contribution and meet the inclusion criteria.

Twenty-one frameworks are classified as being ‘in use’. There is clear evidence for five frameworks being discontinued due to the institutional owner using different frameworks, while six frameworks are categorised as being in ‘limited use’ due to lack of new results being produced in recent years (some of which may have been discontinued). Also, for some frameworks categorised as being ‘in use’, it is unclear how some of them are currently being used. Seven frameworks are still under development or in the finalisation stage (‘pilot test’ in Table 1). Several frameworks were created more for internal planning purposes than for external use (see 4th column in Table 1), or they have not yet been publicly released due to being recently created. UNICEF and the World Bank are the agencies with the most frameworks—at six and five frameworks, respectively—followed by Eawag and WHO with three each, and six organisations with two each (see Table 1).

In terms of the scope of frameworks, twenty cover multiple aspects of the policy enabling environment (of which four are CWIS frameworks), ten explicitly target a segment of the enabling environment, while nine are market assessment with a focus on market dynamics. To some extent the latter frameworks capture specific policy variables.

**Table 1 ijerph-22-01868-t001:** Frameworks and tools that capture the sanitation economy.

Tool or Framework Name (Alphabetical Order)	Lead Agency or Initiative	Sector	Status	Scope ^2^	No. of Indicators	Countries Applied ^3^	Level	Frequency	Latest	Approval	Report	Refs.
Accountability, Mandate, Resources	Sanivation	S	In use ^1^	M	20	9	National	One-off	2023	No	Internal	[30]
Barriers to Scaling Up Sanitation Enterprises	Oxford, Eawag	S	Pilot	M	0	20	Enterprise ^5^	One-off	2023	No	Article	[31]
Building Block Frameworks	SWA	WASH	Limited	C	0	>30	National	2–3 years	2022	Yes	Internal	[32]
	IRC	WASH	In use	C	43	9	Multiple	Regular	2024	No	Agency	[33]
	WaterAid	WASH	In use	C	88	12	Multiple	3–5 years	2024	Yes	Agency	[34]
	UNICEF, SIWI	WASH	In use	C	0 ^4^	0 ^4^	Multiple	- ^4^	- ^4^		None	[35]
Citywide Inclusive Sanitation (CWIS)	World Bank	S	In use	C	>80	ND	Project	One-off	-	No	None	[36]
	BMGF	S	In use	C	20	ND	Cities	One-off	-	Yes	Online	[37]
	Eawag	S	In use	C	-	1	Cities	One-off	2022	No	Thesis	[38]
	Sanitation CityScape	S	Limited	C	0	1	Town	One-off	2020	No	None	[39]
Collaborative Behaviours	SWA	WASH	In use	C	18	68	National	4 years	2020	Yes	Agency	[40]
Equiserve	Athena Infonomics	S	In use	T	>15	12	Cities	One-off	2024	Yes	User stories	[41]
GLAAS (UN-Water)	WHO, UNICEF	WASH	In use	C	>100	124	National	2–3 years	2022	Yes	Agency	[7]
Framework for Integrity in Infrastructure Planning	WIN	WASH	Pilot	T	22	10	Project ^6^	One-off	2024	Yes	Internal	[42]
Investment Cases	SHF	S	Discont.	M	16	4	National	One-off	2023	No	Agency	[43,44]
Investment Climate for Waste Reuse	IWMI	S	In use	T	5	15	National	One-off	2023	No	Article	[45,46]
Market-Based Sanitation Indicators	WASHPaLS #2	S	Pilot	M	13	ND	Subn.	One-off	2023	No	None	[47]
Market-Based Sanitation (Guidance on)	UNICEF	S	In use	M	35	ND	Subn.	Periodic	ND	No	None	[48]
Market-Based Sanitation Pre-conditions Assessment	iDE	S	Pilot	M	27	3	Towns	TBD	ND	No	None	[49]
Market Driven Approach for FST Products	Eawag	S	Limited	M	4	5	Cities	One-off	2015	No	None	[50]
Market System Resilience Index (MSRI)	iDE	S	Limited	M	39	9	National	Annual	2024	No	None	[51]
Policies, Institutions and Regulations	World Bank	WSS	In use	C	0	10	National	One-off	2022	No	Agency	[52]
Principles on Water Governance	OECD	WR, Water	In use	C	36	ND	National	Periodic ^1^	2024	Yes	Agency	[53,54,55,56]
Regulation For Inclusive Urban Sanitation	ESAWAS	S	In use	T	32	8	National	Periodic ^1^	2023	No	Agency	[57]
Sanitation and Health Guidelines	WHO	S	In use	C	0	ND	Multiple	One-off	2020	No	None	[58]
Scaling Up Rural Sanitation (SURS)	World Bank	S	Discont.	C	45	13	National	Annual	2016	No	Agency	[59]
Scorecard for Investment in Water Security	OECD	WR, WASH	Pilot	T	29	7	National	Periodic ^1^	2023	Yes	Agency	[60]
SDG 6 Global Acceleration Framework	UN-Water	WR, WASH	In use	C	0	0	Multiple	-	-	-	None	[61]
Sector Functionality Framework	WSUP	WASH	Limited	C	21	6	National	Periodic	2018	No	Agency	[62]
Service Delivery Assessment (and CSOs)	World Bank	WASH	Discont.	C	27–45 ^7^	46	National	One-off	2015	Yes	Agency	[63,64,65,66,67]
Stargazer Framework	PSI	WASH	Pilot	M	25	10	National	TBD	ND	No	None	[68]
Sector-Wide Sustainability Check	UNICEF	WASH	In use	C	>100 ^8^	17	Multiple	Periodic	2025	Yes	Agency	[69]
WASH Sustainability Index Tool (SIT)	USAID, Rotary International	WASH	Discont.	C	>200	12	Multiple	One-off	2012	No	Agency	[70]
WASH Action Group Indicator Framework	e-MFP, Aqua for All	WASH	Pilot	T	20	0	SME and FI	TBD	2024	No	None	[71]
WASH Bottleneck Analysis Tool	UNICEF	WASH	In use	C	>100	>50	Multiple	Periodic	2024	Yes	Agency	[72]
WASH Poverty Diagnostics	World Bank	WASH	Discont.	T	0	18	National	One-off	2018	No	Agency	[73]
WASHREG	SIWI, UNICEF, WHO, IADB	WASH	In use	T	27	5	National	Periodic	2025	No	None	[74]
Water Integrity Risk Index	WIN	WASH	Limited	T	7	12	Community	One-off	2019	No	Agency	[75]
Water Investment Scorecard	AIP-PIDA	WR, WASH	In use	T	47	12	National	Periodic	2023	Yes	Internal	[76]

Abbreviations: For agency names, see Abbreviations. WR—water resources. WASH—water supply, sanitation and hygiene. WSS—water supply and sanitation. ND—no data. NA—not available. HLM—SWA high-level meetings. e-MFP—European Microfinance Platform. SME—small and medium-size enterprise. FI—financial institution. FST—Faecal Sludge Treatment. Explanation of column headings. ‘Status’: In use—clear evidence of framework still being used (in 2025). Limited use—no recent applications but no clear evidence of discontinuation. Pilot—framework undergoing initial application prior to roll out. Discontinued—clear evidence of institution no longer intending to use the framework. ‘Scope’: either attempts to cover enabling policy environment comprehensively (C) or explicitly targets a segment of the enabling environment (T), or is a market assessment with a focus on market dynamics (M). ‘Approval’: either governments supply the information for the framework or evidence that governments have approved the results. ‘Report’: results are available either in a public agency report, an academic publication or an internal document. Other notes: ^1^ Mainly for internal use or not public due to pilot phase. ^2^ Focus: M—market strengthening focus. C—comprehensive enabling environment. T—targeted enabling environment. ^3^ Where possible, this number reflects until mid-2025. ^4^ See WASH BAT, which used and adapted the SIWI/UNICEF framework. ^5^ 36 enterprises in 20 LMICs; ^6^ 10 large infrastructure projects in one Latin American country. ^7^ Around 3–5 indicators for each of the 9 building blocks, varied by country. ^8^ Includes 55 rural and 78 urban indicators, with many similarities between them.

### 3.2. Evolution of Sanitation Economy Frameworks

This section introduces the tools in chronological order to show the evolution in thinking in systems analysis, and to explore the extent to which later frameworks (and tools) have built on the structure, successes and failures of earlier frameworks (and tools). This knowledge serves as a foundation for later sections. In particular, it is noted here the shift from understanding the overall enabling environment in earlier frameworks to later frameworks drilling down into understanding specific aspects more deeply such as regulation, finance and market dynamics.

One of the first identifiable frameworks was developed in the mid-2000s by the World Bank’s Water and Sanitation Program (WSP) in the Scaling Up Rural Sanitation (SURS) programme, where eight sector building blocks were identified as being necessary to promote sanitation acceleration at national and sub-national levels [21,59]. The building blocks covered mainly public functions (policy, monitoring, funding, etc.) but also focused on the availability of implementation capacity and sanitation products in the marketplace. Initially, the framework was used to help identify the weakest areas of the enabling environment that needed strengthening, with ongoing monitoring over several years to assess progress. Later it helped answer whether strengthening these building blocks had led to the envisaged progress in sanitation coverage [59]. The framework was also used to explain to donors how WSP’s support to the enabling environment had helped increase coverage and to count attributed beneficiaries.

In the same period, the World Bank developed an analytical method for assessing the water and sanitation enabling environment, termed the ‘Country Status Overview’ (CSO), which was applied in 16 African countries in 2006 [63]. The CSOs presented a snapshot of current service coverage, and they reviewed national strategies, institutional arrangements, sector financing, sector M&E and sector capacity, followed by an examination of institutional and financial sustainability in both rural and urban areas. In 2008, the method was extended to further countries—covering 32 African countries in total—with a focus on sanitation and concluded with eight priority actions for strengthening the enabling environment [64].

In 2008, the World Health Organization piloted the UN-Water Global Analysis and Assessment of Sanitation and Drinking-Water (GLAAS). The first report noted work that the WHO Regional Office for Africa had published in 2000 titled “Water supply and sanitation sector assessment” which used selected country examples to assess the coverage, costs and investments in water and sanitation, as well as policy, planning, institutional responsibilities and capacity [22]. Over successive applications of UN-Water GLAAS, the survey has evolved into a significantly more detailed instrument for monitoring progress in the WASH enabling environment—with four main sections in the survey form covering governance, monitoring, human resources and finance. It has expanded to monitor over 100 LMICs and includes a few high-income countries [7]. In previous cycles, findings were also reported from a survey of 23 external support agencies on their aid priorities and spending. The latest UN-Water GLAAS report was published in 2022 and another one due in 2025.

In 2011, a second round of Country Status Overviews (CSO2) were published for 32 African countries and included a synthesis report [65]. The CSO2 built on the methods and processes developed in the earlier CSOs and presented the ‘CSO2 scorecard’ which contained nine building blocks across three pathways—enabling, developing and sustaining—which are needed to translate finance into services through government systems, and in line with Paris Principles for aid effectiveness. The success of the CSO2 in framing systems development as nine building blocks and three pathways contributed to the decision to expand it to other regions—covering seven Asian countries [66] and seven Latin American countries [67]—where it was called Service Delivery Assessment (SDA). Since these publications, the CSOs and SDAs have been discontinued, as the World Bank’s thinking further evolved (see later).

In 2011, UNICEF convened several global partners to receive inputs to the design of a new tool which would enable a transparent and collaborative process for diagnosing and thereby strengthening the WASH enabling environment. It became known as the WASH Bottleneck Analysis Tool (WASH BAT). In 2012, UNICEF piloted and started rolling out the first version of the WASH BAT as a way for governments and development partners to jointly identify the constraints on WASH progress and propose a costed, prioritised and sequenced plan of action. The tool covered 17 enabling factors touching mainly on public functions to support WASH sector development. Since 2016, the tool structure was updated, and an online version of the tool has been applied in over 30 countries (including sub-national applications in many countries) [77]. The revised framework and criteria align with UNICEF’s WASH sector functions [35,72]. Since 2020, additional assessment criteria provide options to customise the WASH BAT exercise for humanitarian and climate-affected contexts [78].

In 2012, USAID and Rotary International developed the WASH Sustainability Index Tool and piloted it in three countries [70]. The framework has had further applications in at least nine more countries in Africa and Asia, either under contract with the tool owners or adopted by other organisations. The tool influenced later frameworks on sustainability, such as UNICEF’s sustainability check tool [69] and IRC’s WASH sustainability assessment tool [79], which is based on the IRC Building Blocks [33].

In 2015, the World Bank initiated the WASH Poverty Diagnostics, which focuses on the links between WASH, poverty and health, identifying the binding constraints to improving WASH service delivery for poor people [73]. The 18 country reports assess the costs and financing to reach WASH targets, as well as policies and their effectiveness, drawing on UN-Water GLAAS data [7]. The framework adopts three diagnostic lenses to identify constraints and three areas of action: oversight and accountability, intergovernmental arrangements, and capacity.

In 2015, the Organisation for Economic Cooperation and Development (OECD) released its Principles on Water Governance, which provide the 12 “must-dos” for governments to design and implement effective, efficient, and inclusive water policies. To date, they have been endorsed by 37 OECD member countries, 7 non-member countries and 140 stakeholder groups [53]. Several countries have applied the principles, and two regional benchmarking reports have been conducted [54,55,56].

In 2016, the Sanitation and Water for All (SWA) partnership formulated five WASH sector ‘building blocks’ which were prepared by one of SWA’s multi-partner working groups to help identify ‘what’ needs to be strengthened [32]. It was endorsed by the SWA Steering Committee. The SWA Building Blocks have proven to be useful in bringing a common understanding amongst stakeholders and provide a basis for monitoring and analysis. The Building Blocks have been scored by countries using a traffic light system as part of the preparation process for the SWA high-level meetings.

Since 2016, several international agencies have adapted the SWA building blocks to meet their own specific needs, including IRC [33], WaterAid [34] and Stockholm International Water Institute (SIWI) and UNICEF [35]. The Building Block approach has been promoted through platforms such as Agenda for Change, of which IRC and WaterAid are members. Also, the SWA Collaborative Behaviours—which measure the ‘how’ of WASH sector development with a focus on strengthening government leadership, using country systems, and increasing coordination across development partners—have been scored for most SWA member countries in 2016 and 2020 [40].

In 2016, a group of development partners identified ways to accelerate progress in providing sanitation services for the urban poor, resulting in the City-Wide Inclusive Sanitation (CWIS) concept and “Call to Action” [80] and later described [27,81]. Since then, CWIS has become an umbrella term which has been taken on by many organisations and has developed in many directions. Four frameworks were selected for this review to provide an indication of the conceptual advances and general direction of diagnostic assessments in CWIS initiatives [36,37,38,39]. Prior to CWIS, others such as the International Water Association (IWA), Swiss Federal Institute of Aquatic Science and Technology (Eawag) and the German Agency for International Cooperation (GIZ) had worked on city-wide planning approaches [24,29]. These approaches are planning methodologies and do not themselves contain indicators for tracking over time, hence are not reviewed in this paper.

In 2016, the World Bank initiated new work on WASH sector governance [82] which led to a more developed version of the framework in a publication “Policy, Institutions and Regulations” [52]. This publication provides an assessment framework under several major blocks, but the framework does not include indicators.

The multitude of frameworks and sector monitoring initiatives described above testify to the growing importance given by international and national agencies to the WASH enabling environment. These have led to the generation of a significant array of data sets and reports, and a more nuanced understanding of the role of the enabling environment in WASH sector development in what is broadly termed ‘systems thinking’. However, given the many complex interactions that need to occur to strengthen the enabling environment and the many diverse contexts in which the enabling environment still needs to be strengthened, understanding of the WASH enabling environment is far from complete [25,28]. Several initiatives described above have been discontinued or have not reached any degree of scale. Some frameworks are broad in their scope, and they often lack the ability to enable in-depth analysis and understanding. Also, insufficient studies have been conducted to analyse the data available to understand the specific drivers of sector progress. No frameworks have yet attempted a summary score or ranking of countries based on the strength of their enabling environment or investment readiness.

Possibly for these reasons, later frameworks, mainly since 2016, have chosen to deepen a specific aspect of the enabling environment, or conduct analyses to understand sustainability, market status, market attractiveness and investment readiness. For example, one aspect addressed by several tools since 2023 is financing, and how to attract more investment for WASH, or water resources more broadly. These include the OECD’s ‘Scorecard’ which assesses the enabling environment for investment in water security [60], the Continental Africa Water Investment Programme’s (AIPs) Water Investment Scorecard [76], the Sanitation and Hygiene Fund’s Investment Cases [43,44], and the International Water Management Institute’s Investment Climate Tool [45,46]. The European Microfinance Platform (e-MFP) and Aqua for All developed a structured set of indicators to guide the activities of impact investors in the WASH sector [71].

Several new tools have focused on market strengthening specifically since 2020, including UNICEF’s Guidance on Market-Based Approaches [48], Eawag’s Market Driven Approach for the selection of Faecal Sludge Treatment Products [50], Sanivation’s Accountability, Mandate and Resources framework [30], International Development Enterprises’ (iDE’s) Market System Resilience Index [51], WASHPaLS #2 Market-Based Sanitation Indicators [47], iDE’s Market-Based Sanitation Pre-conditions Assessment [49], Oxford University’s Barriers to Scaling Up Sanitation Enterprises [31], and Population Services International’s (PSI) Stargazer Framework [68]. e-MFP and Aqua for All have led the development of a set of indicators to guide the activities of impact investors in the WASH sector [71]. Several of these frameworks are still under development and/or not yet publicly available.

One aspect that has been the subject of increasingly attention from WASH sector organisations in the past 15 years is that of sustainability [29]. The common occurrence of household toilets going out of use and populations returning to open defecation also came into focus following widespread implementation of Community-Led Total Sanitation (CLTS) campaigns [83,84,85]. Since the early 2010s, several frameworks have been developed, and others expanded, to enable deeper analysis of the determinants of sustainability and to support improved policies and programme selection. The first initiatives include those of USAID [70] and BMGF (e.g., Triple-S implemented by IRC), which provided early learning on the subject. UNICEF initiated its work on sustainability checks in the late 2010s, initially more focused on the project and service level, later expanding to encompass sector-wide aspects [69]. Also, IRC’s building block framework [33] has been extended to include indicators relevant to sustainability [79].

Other newer initiatives have focused on regulation and integrity. These include WASHREG from SIWI, UNICEF, WHO and IADB [74], the Eastern and Southern Africa Water and Sanitation Regulators Association’s (ESAWAS) Regulation Strategy and Framework for Inclusive Urban Sanitation [57], the Water Integrity Network’s (WIN’s) Water Integrity Risk Index [75], and WIN’s Framework for Integrity in Infrastructure Planning [42]. Sanitation planning frameworks include the CWIS frameworks described earlier [27,28,36,37,38,39] and the ‘Equiserve’ tool developed by Athena Infonomics [41] which is under implementation by several international agencies. Equiserve goes beyond the typical variables included in most other planning tools by incorporating city-level sanitation market performance assessment and policy variables. It helps authorities identify investments (and projects) that are needed to cost-effectively increase sanitation coverage, within financing constraints and with a focus on poor households. It is included in this review as it contains several diagnostic elements.

### 3.3. Structure of Frameworks

Of the thirty-nine frameworks that cover sanitation, twenty could be considered to be comprehensive frameworks that attempt to examine the full breadth of issues related to sector governance and the enabling environment. The remaining nineteen frameworks focus on financing or investment (four frameworks), market monitoring and strengthening (nine frameworks), regulation or integrity (four frameworks), poverty assessment (one framework) and planning (one framework) (see Table 1, where the reference for each framework is provided). There are clear similarities and overlaps between the structure and indicators covered by the thirty-nine frameworks, with some market aspects covered by enabling environment frameworks, and some governance aspects covered by market, finance and integrity frameworks.

The twenty comprehensive enabling environment frameworks that utilise sector building blocks are presented in Table 2. All frameworks consist of a clear structure but use different terms to denote the structure, such as building blocks, pillars, dimension, functions, objectives, criteria, principles, outcomes, accelerators, components and assessment areas. These are collectively referred to here as ‘pillars’. Frameworks define between three and thirteen pillars, with an average of 6.5 pillars. While there is some variation between the frameworks in their comprehensiveness, they most commonly include sector policy and strategy, institutional arrangements, regulation, public financial management (financial planning, tracking, auditing), finance (sourcing additional finance), monitoring and review, and capacity development. Also covered by at least half of the frameworks are planning, coordination and harmonisation, accountability, and equity and social inclusion. Themes such as service delivery, government leadership, water resource management, climate resilience and sustainability, infrastructure development, behaviour change and social norms, participation and engagement, decentralisation and political leadership are covered by 3–4 frameworks each. Learning, market availability, innovation and cost-effective implementation are covered by 1–2 frameworks each.

Frameworks differ in the focus and extent of coverage of each theme, with considerable variation in the number of indicators. Several frameworks are analytical frameworks which are not driven by indicators as such (e.g., SWA’s Building Block Framework, World Bank’s Policies, Institutions and Regulations, UN-Water’s SDG 6 Global Acceleration Framework, and World Bank’s WASH Poverty Diagnostics). The WHO’s Guidelines on Sanitation and Health provide an implementation framework for sanitation, while referring the user to existing surveys and monitoring initiatives to populate relevant indicators [58]. Some frameworks have a large number of indicators at over 100 indicators (e.g., UN-Water GLAAS, UNICEF WASH BAT, USAID’s WASH Sustainability Index, and World Bank’s Service Delivery Assessment). Other frameworks use less than 50 indicators (e.g., IRC’s Building Blocks, WSUP’s Sector Functionality Framework, SWA’s Collaborative Behaviours, OECD’s Principles on Water Governance and World Bank’s Scaling Up Rural Sanitation).

The aspects covered by the four frameworks focused on finance and investment are compared in Table 3. Most frameworks cover the water governance, policies, regulations and mechanisms to mobilise water investments. The OECD Scorecard assesses most comprehensively the overall attractiveness of a country for investment (i.e., non-water specific), while IWMI’s Investment Climate also assesses general governance issues. The OECD Scorecard also assesses the capacity of service authorities and providers, and linkages with other economic sectors. AIP-PIDA’s Water Investment Scorecard uniquely assesses the finance sector development, means of enhancing investment performance and trends in expenditure from major sources. IWMI assesses ‘entrepreneur ecosystems’ focusing on markets and business networks. The WASH Action Group includes separate indicators for SMEs and financial institutions, focusing on business, climate, social, and access indicators.

Nine frameworks enable an understanding of the sanitation market, and this includes some aspects of the broader enabling environment. Table 4 maps the key aspects of the market assessments, indicating quite some variability between them. The frameworks with the broadest scope are PSI’s Stargazer Framework, iDE’s Pre-conditions Assessment and WASHPaLS #2 MBS indicators. Some frameworks build on other frameworks. For example, iDE developed their MBS Pre-conditions Assessment drawing on UNICEF’s and WASHPaLS #2’s MBS approaches. Oxford University’s Barriers to Scaling Up Sanitation Enterprises tool focuses on five constraints: financial, regulatory, infrastructural, social, political. The WASH Action Group’s Indicator Framework includes a series of business, social, climate and service level indicators, distinguishing indicators for SMEs and indicators for financial institutions.

Several aspects are common to multiple market-based frameworks, although they may use different terminology, covering consumer demand or market size, market structure and degree of competition, supply chain, policy enabling environment, business enabling environment, financing and market attractiveness. Workforce, training and climate resilient infrastructure are included in at least three frameworks. Social barriers, sanitation coverage (as indicative of demand), inclusion and the broader development context are included in two frameworks. The availability of public goods is unique to WASHPaLS #2 and the nature of the physical environment is unique to the MBS Pre-conditions Assessment.

Four frameworks focus on regulation and integrity. The WASHREG involves a multi-stakeholder approach to identifying national regulation gaps and challenges in water and sanitation services provision [74]. Criteria cover service quality, consumer protection, tariff setting, competition, environment and public health, and the assessment results in a set of proposed practical actions to further develop, strengthen and align regulatory roles and responsibilities. The ESAWAS framework has a regulation lens, examining sanitation definitions, policies, technologies available, the legal framework and regulatory instruments to support inclusive urban sanitation [57]. It examines the roles and responsibilities of key institutions and of different players along the sanitation chain. In this sense, it dives into more detail in the ‘Institutional set-up’ pillar covered in the broader enabling environment frameworks. The Water Integrity Risk Index from WIN includes three types of risk: investment integrity risk, operations integrity risk and client-utility interaction integrity risk. Each of these includes public procurement risk indicators [75]. The Framework for Integrity in Infrastructure Planning (FIIP) was developed by WIN and the Infrastructure Transparency Initiative (CoST), with support from the Inter-American Development Bank [42]. It identifies and scores seven risks related to integrity across the project planning cycle, covering decision-making, conflicts of interest, and misuse of public funds.

Two frameworks have their own category. The Equiserve Tool is a planning framework for urban sanitation service providers and is included as it includes diagnostic indicators [41]. It identifies current sanitation service coverage and assesses the costs and means of achieving coverage targets. The tool has a poverty angle, analysing poor and non-poor households differently and the affordability of WASH services. The opportunity and constraints faced by service providers in expanding services across the city and to the poor specifically are assessed, and a financial analysis conducted. The World Bank’s WASH Poverty Diagnostics examines the poverty status of those without WASH services, the WASH-health linkages, and what barriers poor people face to accessing WASH [73]. It conducts a financial gap analysis and analyses sector oversight and accountability, intergovernmental arrangements, and capacity.

### 3.4. Application of Frameworks

This section answers questions related to which countries and sub-national contexts the frameworks have been applied in, the data collected (scope, frequency) and the robustness of the data (see Table 1). However, despite being informed by a focal point in the institutional home of the frameworks, it may be impossible to know all the places where a tool or framework has been applied or used, and this analysis is only based on the information provided by focal points and available literature.

The framework with the most widespread application and over the longest period is the UN-Water GLAAS, which was piloted in 2008 and has been implemented in regular cycles in 2010, 2012, 2014, 2017, 2019, 2022 and 2024. In the latest report in 2022, data were reported by 124 LMICs [7]. The GLAAS is the broadest framework with the most amount of data generated, collecting data across water, sanitation and hygiene sub-sectors for households, schools and healthcare facilities. The data are national, with a rural/urban breakdown for many of the questions. There is a lag period of about one year between data collection and reporting, although for some official data such as financial data, the lag period can be more than two years. The questionnaire is filled in by relevant government ministries and often involves meetings and workshops with sector development partners for data collection and data validation purposes. WHO, UNICEF and other agencies provide a supporting budget for a considerable number of countries to enable them to hold meetings or workshops. The data are endorsed by the submitting governments, and they are publicly available on WHO’s website and in the synthesis report [7].

Frameworks also with extensive national application in at least forty-six countries each are SWA’s Collaborative Behaviours and Building Blocks, UNICEF’s WASH Bottleneck Analysis Tool, World Bank’s Service Delivery Assessment and OECD’s Principles of Water Governance. In the second round of SWA’s Collaborative Behaviour country profiles, 68 countries were assessed in 2020 [40]. The assessment draws on UN-Water GLAAS data and other available sources, hence are largely based on approved data sources. The SWA does not plan to continue publishing updates to the profiles. SWA’s Building Blocks have been used for self-assessments by countries using a traffic light scoring methodology prior to several SWA high-level meetings. Their main influence has been in motivating other agencies to further develop their own frameworks using the Building Blocks, for example, IRC, WaterAid, and SIWI and UNICEF.

The World Bank’s Country Status Overview (in Africa) and Service Delivery Assessment (in Asia and Latin America) were conducted by consultants and in-country staff, with consultation workshops held for governments and partners to provide feedback [63,64,65,66,67]. In one country, India, SDAs were conducted at sub-national level in four States. The last SDA report was published more than ten years ago, with the World Bank’s approach to assessing the enabling environment evolving into other initiatives.

UNICEF’s WASH Bottleneck Analysis Tool has been applied in at least fifty countries since 2012 for urban and rural areas separately and in water, sanitation and hygiene sub-sectors [77]. In most countries it has also been applied in selected jurisdictions at sub-national level, although in Bangladesh it was applied in all eight Divisions. In a few instances it has been applied at municipal levels. The >100 indicators per sub-sector are not scored quantitatively, rather they are assessed on a 5-point scale through discussion with a wide range of stakeholders in a workshop setting. The tool’s purpose is to identify the major bottlenecks and formulate a sequenced, costed and prioritised plan to remove the bottlenecks [72]. It is typically endorsed by the government, and a report made available on the UNICEF’s WASH BAT website. Some countries have applied the WASH BAT a second or third time [77], and UNICEF reports that at least 10 countries use the WASH BAT each year, although repetitions are not needed frequently given the multi-year action plan that is identified through its first application.

OECD’s Principles on Water Governance have been applied in detailed national studies in at least five countries from 2017 to 2022 (Argentina, Brazil, Korea, Peru and South Africa) and in one of these (Brazil) it was applied twice, five years apart [53]. Furthermore, OECD conducted benchmarking studies based on a survey on water governance comprising 46 questions, which were answered using secondary data and information obtained from a number of sources. Consequently, the report “Water Governance in Asia-Pacific”, published in 2021, provides a regional analysis of the state of play of water governance in 48 countries of the Asia-Pacific region [54]. In the same year, a similar report for Africa was published “Water Governance in African Cities” which assessed water governance dimensions across 36 African cities in 20 countries [55].

Three other frameworks from the World Bank have been applied in between ten and eighteen countries at national level. The Scaling Up Rural Sanitation framework scored 45 indicators across eight pillars through an internal assessment, whose main purpose was for internal monitoring and planning and therefore received limited government endorsement [21]. The exercise was conducted annually in thirteen countries that were the focus of the global scaling up rural sanitation initiative of the Water and Sanitation Program, until 2016 when the programme was discontinued [59].

The WASH Poverty Diagnostics was the next flagship of the Water and Sanitation Program, an initiative that was applied in eighteen countries [73]. It uses a common analytical methodology customised to the specific needs and demands in each country, producing new ways of examining WASH coverage through a poverty lens (targeting the poorest 40 percent of the population, termed ‘bottom 40 percent’) and recommending key policy actions to accelerate progress for the poor. World Bank staff and consultants collect the required information together with sector partners. Financing data are drawn from the UN-Water GLAAS 2017 report.

The Policies, Institutions and Regulations framework is an assessment methodology with no quantitative indicators that was applied in ten countries through rapid assessments led by the World Bank. The report was structured by several Building Blocks and made a series of recommendations based on observations in the ten countries [52].

The remaining frameworks have been applied in as many as fifteen countries, and in few instances have they been applied more than once in a country (see Table 1). Several frameworks are still being used, and some have yet to be fully scaled up. Some frameworks are still in their initial deployment phase, and a greater number of countries are expected to be covered in the years ahead, such as the OECD’s Scorecard [60], iDE’s MBS Pre-conditions Assessment [49] and AIP-PIDA’s Water Investment Scorecard [76].

Sub-national assessments have been implemented by several enabling environment frameworks. The IRC Building Block analysis has been implemented at both national and district levels covering nine countries [86], while the WASH Bottleneck Analysis Tool has been applied at multiple levels across at least 50 countries, from national down to city level, depending on government demand [77]. Although the UN-Water GLAAS survey is completed at national level, it collects information on some indicators relating to policy implementation at sub-national level [7]. While the CWIS Initiative is being implemented in an increasing number of cities and countries, information is not readily available on the full extent of application given its widespread use.

The credibility of data is of paramount importance to gaining the buy-in of framework audiences. First, the indicators should be clear, with a scoring methodology (if on a scale) and definitions of terms provided. Second, given that much of the data relies on available government or donor reports and data sets, the data sourced should not be outdated, i.e., no longer relevant. Third, to validate the data and reflect a range of perspectives, there should be opportunity for consultation or peer review from stakeholders. This is particularly important for several frameworks given they often involve assessments made by individuals or in groups where the perceived strength of accomplishment is assessed and scored (e.g., Likert scale or traffic light system) [32,33,34,35,72]. A final step of a stakeholder validation process may be the formal approval by a mandated institution, normally a government agency.

Some of the frameworks that involve assessments by an individual or groups and that are later peer reviewed or validated, and have some form of government verification process, endorsement or approval include: the UN-Water GLAAS survey [7], the WASH Bottleneck Analysis Tool [72], the Equiserve tool [41], the AIP-PIDA Water Investment Scorecard [76], OECD’s Principles on Water Governance [53], the UNICEF Sector-Wide Sustainability Checks [69], and the SWA Building Blocks and Collaborative Behaviours [32,40]. If a quantitative verifiable data point is not available, these assessments provide a score on a scale, with each point of the scale being defined. For example, the UN-Water GLAAS assesses a strategy or plan not by its presence or absence alone, but also whether it is officially approved, up-to-date, evidence-based and specifically addresses equity concerns [7]. Some frameworks provide less clear scoring criteria, such as the SWA Building Blocks [32], the WASHREG [74] and the UNICEF WASHBAT [72], which assess the performance of a particular criterion or aspect on a scale.

OECD’s Principles on Water Governance framework requests respondents to signal the level of consensus among stakeholders in order to reflect the diversity of opinions during the discussion [53]. The OECD tool also asks if changes are expected in three years’ time on water governance performance, with three response options: ‘improvement’, ‘stable’ and ‘decrease’. iDE’s MRSI includes ‘determinants’ rather than indicators, and provides guidance with definitions, data sources, and interview questions and how they should be assessed/scored [51]. WSUP provides definitions for each indicator, which are shared with workshop participants and market actors for the assessment exercise [62].

Some frameworks rely on data compilation initiatives outside the WASH field, and include financial data (e.g., GDP, public expenditure), overseas development assistance (OECD’s creditor reporting system), and household surveys (e.g., censuses, health or child indicator surveys). As the data are reported by international organisations or national bureaus of statistics, there is assumed to be overall buy-in and acceptance of the figures by governments, although this will not always be the case. Some frameworks draw on composite indices compiled by international organisations. For example, the OECD Scorecard [60] draws on the World Bank’s World Governance Indicators, which aggregates data from more than 30 think tanks, international organisations, nongovernmental organisations, and private firms across the world. AIP-PIDA’s Water Investment Scorecard [76] draws on financial sector strength information from the African Development Bank’s Country Policy and Institutional Assessment (CPIA) Index and uses S&Ps Sovereign Risk ratings. IWMI’s Investment Climate [45,46] draws heavily on the World Bank’s Doing Business framework. Hence, the frameworks conclusions are as robust as the composite indicators and reported data on which they draw.

Having repeated applications of the frameworks are valuable to enable monitoring, tracking of progress and evaluation. However, few frameworks have been applied repeatedly over a longer time period. Only one tool—the UN-Water GLAAS—has a global scope that covers most LMICs and is applied every 2–3 years in each country, implemented in seven cycles so far, thus enabling some tracking of progress across countries using a standardised methodology [7]. However, with some adjustments to the GLAAS country questionnaire each cycle, the ability to track all indicators across all cycles is reduced. The UNICEF WASH BAT has been applied in several countries more than once (Madagascar, Nigeria, Pakistan) or gradually extended over time to more sub-national levels or cities (e.g., Iraq, Peru) [77]. Also, IRC reports time series data on the changing building block scores since 2017 in a dashboard and it also reports scores for several countries in Annex 2 of its Annual Reports: Burkina Faso, Ethiopia, Ghana, India, Niger, Uganda [86]. WaterAid plans to repeat the WASH system building block assessments in 2027–28 to understand how the system has changed during the 5-year WaterAid Country Programme strategy period [34]. However, as many as 20 of the frameworks appear to have been implemented as a ‘one off’ exercise, with no wider application in more countries or repeat exercises in case study countries (see Table 1).

### 3.5. Dissemination and Use of Framework Results

Frameworks achieve impact through a combination of different activities, although there is no blueprint for success; and impact per se—such as increased sanitation investment or coverage—is difficult to attribute to the use of a framework. These activities are described in four steps below. In the first instance, it is important that a report is made available to key stakeholders and target users of the key findings and recommendations. The methodology should be clearly described, and the scoring and interpretation should be transparent. Second, stakeholders and users should formulate a plan, or adapt existing plans, based on the results and recommendations. Third, there should be monitoring and follow-up of actions. When these three actions are implemented effectively, impact may be achieved such as sanitation expenditure increases and quality or coverage of sanitation services are improved. Finally, an evaluation helps understand whether the theory of change is fit for purpose or needs to be changed. One litmus test of the utility of a framework is that key stakeholders—especially government ones—demand that it continues to be used (and not imposed), leading to its repeated application.

Frameworks report data in a variety of ways. The majority of frameworks reporting results use standard tables and graphs, accompanied by text. Some frameworks use data visualisations such as traffic light scoring (e.g., SWA Building Blocks, UNICEF WASH BAT), while some use a combination of data visualisations, tables and interpretative text (e.g., UN-Water GLAAS, WSUP Sector Functionality, World Bank SDA, WaterAid Building Blocks). Frameworks collecting data in multiple countries have used maps with colour-coding to visualise differences between countries and regions (e.g., UN-Water GLAAS, World Bank WASH Poverty Diagnostics). Fewer initiatives provide colour-coding for multiple administrative levels within a country—e.g., the World Bank Scaling Up Rural Sanitation and World Bank Poverty Diagnostics. As well as standard agency reports, some initiatives use InfoGraphics to communicate the main results and messages (e.g., UN-Water GLAAS). The Equiserve tool presents user stories from selected cities where it has been applied. WASHREG publishes ‘action sheets’ which provide details and country examples on each of the six assessment criteria [87].

The transparency, availability and dissemination of publicly available results vary considerably. In general, frameworks are clear on their scoring methodology and have background documents which can be shared on request. Nineteen of the frameworks have presented at least one set of country results in an agency publication and/or an online dashboard which is publicly available (see Table 1). Three frameworks published an academic article or thesis instead of an agency report. Nine frameworks have been applied but the results are not (yet) publicly available, although the results of two of these (the AIP-PIDA Water Investment Scorecard and the SWA Building Blocks) are available to in-country stakeholders. An additional eight frameworks have not made their results publicly available, either because they are in pilot phase (WASHPaLS #2, iDE Pre-conditions Assessment, iDE MSRI, PSI Stargazer framework), or because they have been produced primarily for an internal audience (e.g., Sanivation), or the framework exists mainly for others to adopt (e.g., UN-Water SDG6 Accelerator Framework, SIWI and UNICEF Building Blocks; and Sanitation CityScape).

Several frameworks have dedicated websites which are widely shared: the UN-Water GLAAS reports and data sets can be accessed on dedicated pages either via the UN-Water or WHO websites; the UNICEF WASHBAT, IRC, WaterAid, OECD Principles, Athena Infonomics CWIS, Equiserve, WASHREG, and WIN’s WIRI all have dedicated internet landing pages for their frameworks. Other frameworks can be found on the resources or publications page of the organisational owners, such as World Bank frameworks, SHF, OECD Scorecard, ESAWAS, and WSUP Sector Functionality. Several frameworks are also publicised and available on WASH knowledge portals, such as IRC WASH, Sustainable Sanitation Alliance, Rural Water Supply Network, and the Agenda for Change initiative.

It is rare that a framework report is co-badged with the governments of the countries they have collected data on. Some workshops reports from the UNICEF WASH BAT are co-badged with the governments, as are the UNICEF Sector-Wide Sustainability Checks [69]. Other frameworks are co-badged with other international organisations (e.g., the World Bank’s CSOs and SDAs are co-badged with regional development banks, UNDP and others).

Aside from publications, some initiatives have used international conferences and workshops and national events to publicise their results to increase uptake by governments and development partners. It was not possible to systematically capture these for the thirty-nine frameworks. For major multi-country initiatives led by international organisations, international events and national workshops have been used to consult with partners during framework development, implementation and dissemination. These include the UN-Water GLAAS, OECD Principles, the World Bank SDA and WASH Poverty Diagnostics, the UNICEF WASH BAT and WASHREG. The ten national reports generated by the AIP-PIDA scorecard—and the progress to address the gaps—have been presented to national and international audiences. Workshops, launch events and sessions at international conferences are therefore key moments to sensitise audiences and receive valuable feedback on how to use the results and how to scale up implementation.

Few frameworks have conducted a review of their impact. Some frameworks have been implemented in the context of ongoing programmes of international organisations—some of which have important supporting budgets to strengthen policy implementation—and have led to some uptake by local decision makers. However, an explicit theory of change for the frameworks is generally not publicly available, and evidence is lacking on the extent of uptake and the impact. For some, the theory of change can be guessed from the way the framework was implemented. For example, the World Bank’s WASH Poverty Diagnostics was implemented in 18 countries from 2015 to 2018 and it informed the national dialogue on how to scale up WASH services and make them more available to poor households [73]. It also informed the World Bank’s engagement with countries. However, no documentation is publicly available that describes the process or the impact.

There has been mixed uptake of other frameworks and associated tools. In implementing the sector functionality framework in cities in six countries, WSUP contributed to evidence-based planning and coordination amongst partners [62]. WaterAid is in the process of drawing lessons from its building block assessment, with a focus on the process of implementation [34]. Feedback from users of USAID’s WASH Sustainability Index Tool [70] from applications in nine countries indicates some but limited use of the findings in local discussions, whereas ‘management memos’ which were produced in some applications proved to be more useful [88]. The Water Integrity Risk Index, which was used in 12 communities across seven countries between 2012 and 2019, has been stated by the Water Integrity Network to have not been significantly taken up by decision makers [75].

A framework which has been reviewed in detail is UN-Water GLAAS, although the reviews have limited public availability such as donor reviews. The GLAAS 2023–2030 states that GLAAS activities aim to achieve two outcomes: national monitoring systems for WASH are strengthened, and decisions by governments and development partners are informed by easily accessible data on WASH systems [89]. As part of the GLAAS 2021/22 cycle, WHO sought feedback from countries from a feedback form [90]. While use of GLAAS within a country depended on the status of their WASH sector, several areas emerged as aspects that the GLAAS has strengthened: sector coordination; formulation of policies, plans, regulations and programmes; advocacy for funding/financing for WASH; national monitoring and review systems; international and regional reporting [90].

UNICEF published a review of 5 years of implementing version 2 of the WASH BAT from 2016 to 2020 [77]. During this period, 58 WASH BAT workshops—which is a key stage in the WASH BAT process—had been completed in 32 countries. The review found that most of the stakeholders interviewed—whether they were moderators, facilitators, UNICEF staff, government staff or other WASH sector stakeholders—agreed the WASH BAT workshop and process created the ideal environment for a structured and systematic discussion of the key bottlenecks hindering progress, and what can be done to resolve them. There is evidence that the workshop outputs led to positive outcomes in the countries following implementation of the WASH BAT’s action plan. In the majority of countries, a range of outcomes can be traced back to the WASH BAT process. Findings suggest that the WASH BAT might be less successful in contexts where government and other institutions have low capacity and poor coordination. Applications in contexts experiencing conflict or other political fragility makes prioritisation more difficult. Findings suggest that more successful WASH BATs are found in contexts where the WASH BAT has been adapted to the needs of the country, and integrated into national processes, directly feeding into a programme or strategy.

The SWA Building Blocks and Collaborative Behaviours have had quite extensive use at country level and have influenced others to adopt similar language and develop or refine other Building Block frameworks, such as SIWI and UNICEF, IRC and WaterAid. Starting at the Sector Ministers’ Meeting in March 2016 in Addis Ababa, Ethiopia, the SWA Building Blocks have helped structure the SWA High-Level Meetings and have been used to prepare countries for these meetings and identify some of the key issues to address [32]. To date, however, there is no publicly available document formally assessing the impact of the SWA Building Blocks framework or the country profiles.

The SWA Collaborative Behaviours were developed in 2016 and have undergone two formal processes of developing Country Profiles [40]. The first edition in 2016–17 was released at High-Level Meetings in 2017 and led to the refinement of the framework. The second edition in 2020 generated country profiles for 68 countries and these were released to countries to support national dialogue. Published after the first edition of country profiles, a guidance document recommended five major uses: deepen sector analysis; develop a common vision and agenda for development effectiveness; agree on intermediate steps to be taken by all actors to achieve the jointly agreed vision and agenda; regularly review progress; and take corrective actions [91]. To date, there is no publicly available document formally assessing the impact of the SWA Collaborative Behaviour framework or the country profiles.

EquiServe aims to help shape investments and strengthen city systems to advance equity in sanitation services, and five user stories are provided on its website from cities where it has been applied [41]. The user stories essentially describe how the tool identified key needs and how those needs were addressed by using the EquiServe tool. Its impacts include motivating updates to city master plans, justifying new tariff levels to the regulator, developing monitoring plans, improving investment planning, providing insights to household behaviour, and identifying data gaps, service gaps, and affordable technological options for households. Its initial successes have led to EquiServe’s expansion into additional cities.

## 4. Discussion

The development of (at least) thirty-nine frameworks since the early 2000s indicates the importance of understanding the broader sanitation enabling environment, and a gradual movement away from the preference of international development organisations for vertical programmes that deliver sanitation infrastructure or behaviour change programmes directly. It reflects the often voiced need to adopt ‘systems thinking’ to improve the sustainability of WASH services [25,28]. This shift is evidenced by the significant interest in strengthening the sanitation-enabling environment within a large, diverse global coalition of international development organisations and country governments contained within the Sanitation and Water for All (SWA) partnership. With an intention to increase partners’ commitment to and accountability towards systems thinking, the SWA started implementing the ‘SWA Framework’ in 2018, which incorporates the building blocks and collaborative behaviours [92] and now it periodically monitors the actions committed to under the SWA Mutual Accountability Mechanism [93].

No doubt the significant intellectual capital expended on developing and using different enabling environment frameworks—and the innovation therein—has been vital in creating different experiences and perspectives on how to best measure the enabling environment and market development status. Indeed, this process has helped evolve the understanding on how strengthening the ‘building blocks’ drives sanitation progress, both individually and together. This has led to the more recent development of frameworks which have further expounded specific areas such as finance and regulation to provide an even deeper understanding.

The different pillars and the range of indicators selected by different frameworks demonstrates the diverse objectives of different sector stakeholders. Indeed, it is the mandate of any development agency to conduct their own research on how they can make the greatest contribution to the issue at hand. On the other hand, it has also led to some duplication of effort both across the agencies producing these frameworks and in the use of limited bandwidth of in-country development partners, in particular government staff. In an increasingly resource-constrained ODA environment, resources dedicated to global and national sector monitoring should be used more efficiently, which means avoiding duplication, where possible. When international agencies have shared objectives and similar means of meeting those objectives, it should be agreed how they can best collaborate, including the use of common terminology and the same tools.

Therefore, the question has to be asked: noting the many frameworks that are still being actively implemented, with several new frameworks currently being pilot tested, what is the best way to measure and monitor the sanitation-enabling environment in the future? Also, to improve value-for-money in global and national level monitoring of the sanitation economy, is monitoring best conducted through one single comprehensive framework, or do two (or more) frameworks need to be combined to provide the full picture? The discussion below explores these questions by assessing how existing frameworks address several desirable aspects, covering comprehensiveness, value-for-money, relevance and timeliness, ownership and acceptance, user friendliness, and resourcing.

Comprehensiveness: The majority of frameworks either focus on the broader sanitation enabling environment, a specific aspect of the enabling environment, or characteristics of the sanitation market (see Table 1). Of the broader enabling environment frameworks, there is considerable diversity in terms of which pillars or aspects are included, with none including all aspects (Table 2). Therefore, few frameworks, if any, have taken a fully comprehensive approach. Neither do they explicitly evaluate interconnections between factors within the sanitation economy, also a finding of Valcourt et al. (2020) in their systematic review of systems approaches to WASH [25]. Therefore, the results from current frameworks would have to be pieced together to gain an overall understanding of the sanitation economy. This is not currently possible for global monitoring of the sanitation economy, because few frameworks are implemented in the same country at the same time. On the other hand, as the UN-Water GLAAS is implemented in 2- or 3-year cycles in a large number of LMICs, the GLAAS information on the policy and funding environment can be combined with market or investment assessments conducted in individual countries using another framework.

Value-for-money: To be worthwhile, the benefits of implementing a framework need to (significantly) exceed the costs of generating the data and associated reports. To cut costs, many frameworks rely on secondary evidence, either provided by other sanitation frameworks, by data sets and indices produced by international organisations, or other locally conducted research. Frameworks that collect primary data from various sources include government surveys with stakeholder workshops (UN-Water GLAAS), stakeholder workshops alone (IRC and WaterAid building blocks, UNICEF WASH BAT and WSUP sector functionality), consultants hired to collect diverse pieces of information (World Bank SDA, UNICEF MBS, AIP-PIDA Water Investment Scorecard, World Bank WASH Poverty Diagnostics), mixed methods (Equiserve, CWIS initiatives, Sanivation, OECD Principles, OECD Scorecard, and USAID’s WASH Sustainability Index Tool), enterprise surveys (Oxford’s Barriers to Scaling Up Sanitation Enterprises and WASHPaLS #2 MBS) and household surveys (iDE MSRI). Therefore, the majority of frameworks involve quite considerable financial costs to implement in a single country. Ironically, many of these frameworks are collecting similar information. This apparent duplication of effort underscores the importance of conducting a landscape assessment of ongoing monitoring initiatives in a country to assess the value-added of implementing a different framework and of conducting additional data collection.

Relevance and timeliness: Many frameworks are developed for multi-country application in a standardised way (e.g., World Bank SDA, UN-Water GLAAS and OECD Principles) with the aim of selecting pillars and indicators based on how they chart a critical pathway to strengthening the sanitation enabling environment or market. This has the advantage of ‘economies of scale’ and enabling comparative analyses between countries. On the other hand, if there is no flexibility in how the frameworks are implemented, it risks missing some key opportunities to zero in on what specific bottlenecks are hindering progress. The fact that many frameworks have been discontinued suggests that some of them were not properly conceived or pre-tested with their target audiences and had a strong business case. Therefore, before local implementation, it is important to consult with relevant stakeholders on whether the framework is fit-for-purpose, that it meets the informational needs of the users or target audiences, and that the objectives and target audiences are clear. In implementing the WASH BAT, for example, UNICEF and in-country partners assess the need and demand for the WASH BAT and how it might be fine-tuned. The ensuing workshops enabled additional criteria to be assessed and others deprioritized [72,77]. Frameworks that can be applied at multiple administrative levels have the advantage of generating information for a range of stakeholders and pinpointing bottlenecks, but they may lack specificity unless indicators are refined at each level. This was achieved to some extent when IRC and WaterAid applied their building block frameworks at sub-national levels. Also, to remain relevant, there should be a mechanism to update indicators whose value might change every 1–2 years, while not necessarily involving a complete repeat implementation of the framework. Few, if any, frameworks have achieved this, except the UN-Water GLAAS, which has been implemented every 2–3 years across a large number of LMICs, and by IRC who conducts annual updates for its focus countries.

Ownership and acceptance: Given that sanitation is a public (as well as private) good, it is unlikely to advance without strong government engagement and support. Therefore, it is of key importance that the government is involved at an early stage of framework application in a country. If results can be officially recognised, it helps in the implementation of recommended actions and brings donors on board. Given the multitude of opportunities for technical assistance offered to governments, they should be free to choose what role to play in the implementation of a framework—as owner, leader, contributor, end-user or funder—depending on their interest and capacity. The timing and outputs of the framework results should, where possible, be aligned with government policies, strategies, planning and budgeting cycles, monitoring and evaluation. In Pakistan, for example, the WASH BAT was used prior to the government planning cycle and joint sector reviews at both national and provincial levels [77]. Furthermore, engagement across multiple sectors may be vital to the success of a WASH programme. The UN-Water GLAAS, for example, engaged health, education, regulators and finance ministries to receive their inputs to the survey [7]. The World Bank WASH Poverty Diagnostics included consultation process with various government agencies of 18 countries, as evidenced in the acknowledgement sections of the reports [73]. However, few other frameworks have received such endorsement or strong government engagement and hence have had limited adoption in government processes.

User friendliness and availability: Frameworks should collect and present results in an engaging way. Several frameworks collect data in a workshop setting, which enables an introduction to the framework rationale, explanation of the data needs and consensus building. Others involve a survey to be filled in by focal points. The value of standardisation of indicators and methodologies across countries is that it facilitates easier interpretation by global or regional agencies and enables comparability of results across settings. On results presentation, most frameworks publish lengthy documents with a summary of key findings, while few provide dashboards and enable customisation of results (e.g., by specified indicators or countries). An exception is the UN-Water GLAAS data portal. Therefore, in the future, information technology should be used more to facilitate cross country comparison, simple interpretation, and provision of main takeaways and recommendations. Reports should be easy to access on the agency’s website and on knowledge portals, and should be available with no charge (e.g., some academic publications involve an access charge).

Resourcing of continued framework application: In most cases, it is the organisational owner of a framework who has had to guarantee the budget for its continued application. In only one case has the cost of implementing a framework been published—the USAID SIT which varied from US$44,300 to US$146,000 per country, with the variation partly reflecting the different data collection methods (household surveys, focus group discussions and stakeholder meetings) [88]. This lack of financial cost information reflects the low transparency of development agencies in their cost of operations, but also the difficulty of estimating the full costs when in many cases the major cost item is the time of salaried staff and unpaid time of respondents. In most cases—for the more widely used frameworks—it is an international organisation that prioritises it and allocates general or earmarked donor budget. In only rare cases has a framework been picked up and implemented by a different agency. One example is the UN-Water GLAAS, which has been implemented by the World Health Organization for over 15 years, but which is recently co-owned by UNICEF. The lack of continued organisational support has been a major reason why many frameworks have been discontinued or failed to scale up to more countries or be repeated in a country. In some cases, frameworks have received limited use or gone out of use because the same organisation has created a new framework (e.g., some tools of the World Bank, Water Integrity Network and iDE). Therefore, for continued application over a longer period of time, the framework developers need to develop a strong business case from both the donor and the government perspectives in order to maintain institutional support and to sustain the required budgets for the framework’s continued use.

One key finding that stands out is the lack of formal and public reviews or evaluations of sanitation frameworks. The exceptions where frameworks have been subjected to some form of review on how they have been used are the UN-Water GLAAS [90] and UNICEF WASH BAT [77], although neither of these was fully independent. Some reviews may have been missed due to them being wrapped within a broader review of an organisation’s overall work programme. A constraint faced in designing such a review or evaluation is the challenge of attributing impact to the use of the framework, given that broad sector diagnostic and strengthening efforts are not amenable to experimental methods. Instead, a storyline needs to be developed based on an in-depth understanding gained from local stakeholders on how the framework may have influenced their actions and the actions of others. Resources and rigorous thinking need to be applied so that the sanitation sector better understands the past impact and future potential impact of these frameworks.

## 5. Conclusions

The recognition of the WASH enabling environment’s importance since the early 2000s has led to many diverse frameworks assessing both sanitation market dynamics and the sanitation enabling environment, benefiting nearly all LMICs. At different time points, and with the support of different international organisations, these frameworks have provided valuable evidence to motivate policy makers and financiers to strengthen the sanitation enabling environment and have informed specific actions to do so. However, outside of the UN-Water GLAAS survey, these frameworks have been applied inconsistently, with many discontinued due to insufficient analysis, poor consultation, insufficient budgets, fluctuating organisational priorities, and lack of strong evaluation metrics to assess the tools themselves.

In today’s ODA environment, where funding is increasingly tied to measurable impact, business as usual—fragmented and ad hoc application of market assessment and enabling environment frameworks—can no longer meet the needs of a maturing sanitation market. To unlock private and public investment and accelerate sector growth, frameworks must undergo independent evaluations to ensure they are driving real, sustainable change. Greater collaboration and harmonisation of frameworks are essential to optimise limited resources, build stronger business cases, and unlock the full potential of the sanitation economy, paving the way for long-term market development, increased investment, and measurable progress in global sanitation outcomes.

## Figures and Tables

**Table 2 ijerph-22-01868-t002:** Building blocks or aspects covered by comprehensive enabling environment frameworks.

Building Block or Aspect of the Enabling Environment	SWA ^2^ [32,40]	IRC [33]	WaterAid [34]	SIWI and UNICEF ^3^ [35,72]	CWIS [36,37,38,39]	OECD Principles [53,56]	UNICEF Sustainability Checks [69]	UN-Water GLAAS [76]	UN-Water SDG6 Framework [61]	USAID SIT [70]	WHO Guideline [58]	World Bank PIR [52]	World Bank SURS [21,59]	World Bank SDA [63,64,65,66,67]	WSUP [62]
Sector policy and strategy	Y ^BB^	Y	Y	Y	Y	Y	Y	Y	Y	Y	Y	Y	Y	Y	Y
Legislation and regulation		Y	Y	Y	Y	Y	Y	Y			Y	Y	Y		Y
Institutional arrangements	Y ^BB^	Y	Y	Y	Y	Y	Y	Y		Y	Y	Y	Y		Y
Accountability	Y ^CB^	Y	Y	Y	Y	Y	Y	Y	Y	Y	Y	Y			
Infrastructure development ^1^		Y					Y			Y	Y				
Service delivery			Y	Y			Y			Y	Y		Y		
Coordination and harmonisation	Y ^CB^	Y	Y	Y			Y	Y	Y	Y	Y		Y		
Public financial management	Y ^BB^	Y	Y	Y	Y		Y	Y		Y		Y		Y	Y
Financing	Y ^CB^	Y	Y	Y	Y	Y	Y	Y	Y		Y	Y	Y	Y	Y
Planning	Y ^BB^	Y	Y	Y	Y		Y	Y			Y			Y	Y
Monitoring and review	Y ^BB^	Y	Y	Y	Y	Y	Y	Y	Y	Y	Y	Y	Y		Y
Learning		Y		Y											
Capacity development	Y ^BB^		Y	Y	Y	Y	Y	Y	Y	Y	Y		Y		Y
Government leadership	Y ^CB^		Y	Y											Y
Water resource management		Y	Y			Y	Y								
Climate resilience/sustainability			Y				Y	Y				Y			Y
Behaviour change and social norms			Y	Y			Y				Y			Y	Y
Equity and social inclusion			Y	Y	Y	Y	Y	Y					Y	Y	Y
Participation and engagement			Y			Y	Y	Y							Y
Decentralisation		Y		Y						Y		Y			
Market availability/spare parts							Y			Y			Y	Y	
Cost-effective implementation													Y		
Innovation						Y			Y						

Key: ^1^ Includes asset management. ^2^ Two frameworks from SWA are covered here: ^BB^ SWA building blocks. ^CB^ SWA collaborative behaviours. ^3^ UNICEF’s enabling environment framework which is used in the UNICEF WASH Bottleneck Analysis Tool are covered here.

**Table 3 ijerph-22-01868-t003:** Aspects covered by enabling environment frameworks focused on finance and investment.

Pillar or Aspect	OECD Scorecard [60]	AIP-PIDA [76]	IWMI [45,46]	WASH Action Group [71]
Overall policy framework	Y		Y	
Water governance and policy framework	Y	Y	Y	Y
Regulatory framework	Y	Y	Y	
Capacity of service authorities and providers	Y			
Bankability and sustainability of projects	Y	Y		Y
Affordability and subsidies or grants				Y
Social and environmental inclusion	Y	Y		Y
Trends in expenditure		Y		
Investment and business climate		Y	Y	
Mechanisms to mobilise water investments	Y	Y	Y	Y
Enhancing investment performance		Y		
Entrepreneur ecosystems			Y	
Integration of water in national adaptation plans and climate aspects	Y	Y		Y
Linkages with other economic sectors	Y			

**Table 4 ijerph-22-01868-t004:** Market analysis and market strengthening frameworks.

	PSI [68]	Sanivation [30]	UNICEF [48]	iDE MSRI [51]	iDE Pre-Conditions Assessment [49]	Oxford & Eawag [31]	EAWAG [50]	WASHPaLS #2 [47]	SHF [43,44]
Demand ^1^ and market size	Y	Y	Y		Y		Y	Y	Y
Coverage		Y			Y				Y
Social barriers						Y	Y		
Competition and market structure	Y			Y		Y	Y	Y	
Supply chain	Y		Y		Y			Y	Y
Policy enabling environment ^2^	Y	Y		Y	Y	Y		Y	Y
Coordination	Y								Y
Business enabling environment ^3^	Y		Y		Y	Y		Y	
Public goods ^4^								Y	
Collaboration and connectivity		Y		Y					
Inclusion in planning and delivery	Y		Y						
Financing and market attractiveness	Y	Y	Y	Y	Y	Y	Y	Y	Y
Workforce and training	Y	Y			Y				
Climate resilient infrastructure	Y			Y		Y			
Broader development context ^5^					Y				Y
Physical environment (includes climate)					Y				

Key: ^1^ Includes willingness to pay, stated priority of sanitation and demand activation. For iDE Pre-conditions Assessment includes sufficient household incomes. ^2^ Includes governance, accountability, data systems, political support. ^3^ Includes regulation. ^4^ Public goods include availability of market research, toilet design manuals, training, sales and marketing tools, rural delivery model designs and sanitation loan product designs in the public domain. ^5^ Includes political and economic stability, economic growth, information exposure.

## Data Availability

Further data are available in Annex 1 of the Sanitation and Hygiene Fund report “Review of Frameworks for assessing the strength of the sanitation economy and investment readiness” available at https://www.shfund.org/resources/publications (accessed on 13 March 2025).

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
