# Peer review of "Review of Frameworks for Assessing the Strength of the Sanitation Economy and Investment Readiness"

_ijerph, 2025, doi:10.3390/ijerph22121868_

Round 1
Reviewer 1 Report
Comments and Suggestions for Authors
Feedback on the Review of Frameworks for Assessing The Strength of The Sanitation Economy and Investment Readiness
Overall, a very interesting paper that provides a review of an important aspect of the WASH, and particularly sanitation sector. It has potential to be widely read and well cited. However, some fundamental flaws need to be removed to make this a good scientific paper, since it still largely reads in an agency report format.
- The introduction needs to clarify some fundamental concepts: sanitation economy, enabling environment, framework vs tool etc.
- A clear justification as to the reason for this review. The reasons used in the conclusion, could be brought up here.
- The methodology presented here, misses the analytical part that makes it more transparent and replicable.
- The criteria for inclusion of frameworks needs to be explicit. Even within what is understood as the criteria by reading between the lines, there are easily few important frameworks that are relevant
- Issues within the Tables, particularly Table 1. Missing explanation etc.
- The discussion reads more like an opinion article and missed using the evidence brought in the results.
- The conclusion is not directly integrated to the rest of the article.
Although these are major revisions, I would encourage the authors to make a revised version, since it is an important topic, and most of the hard work has already been done. It would be a great contribution to the sector if this paper could be strengthened and submitted.
Detailed feedback is given below.
Abstract
The research gap, or the need for this particular research objective is unclear in the abstract, esp from the first two statements. Why is an assessment of frameworks necessary? Writing this more clearly would be convincing for a potential reader to move from the abstract to the full paper.
Introduction
- Some deliberation and engagement with what is a sanitation economy, beyond the Cookey definition would be helpful. This is crucial for the entire paper.
- There needs to be some clarity, on which aspects are within the sanitation economy and which aren’t. Is a toilet refreshner part of the sanitation economy?
- Before introducing the topic of private investments in sanitation, it would be useful to have a more systematic understanding on what that entails? What is being referred to as “private investments in sanitation”. Is every household investment then considered. It would be valuable to a reader, if such ambiguity could be removed.
- “In other words, there is an information 85 asymmetry which contributes to the market failure. Therefore, one way to facilitate the sanitation economy and stimulate investment is 87 to make available better information to stakeholders that play a role in stimulating or 88 making investments.” This seems to be the fundamental justification for building the whole paper, but there is very little discussion and citations on this aspect. I would welcome a bit more substantive points to build the justification here. You can also look beyond the WASH sector for such evidence. This is only a suggestion.
- Amjad et al (2015), there is a lot more literature on enabling environment on sanitation than to simply use this one from drinking water. The references from WSUP 2018, Lüthi 2011 etc are directly relevant to your case.
Methods
- I would suggest that the 4 questions are moved from methods to the introduction section’s last part.
- The method section is currently not offering any information on the concrete aspects of how the results were obtained, in terms of the analysis, the interview questions, etc. The idea of methods is for this assessment to be replicable by anyone else. Currently there is not enough information apart from a list with the types of methods that were used.
Results
- Where can we see the “evidence” for use or discontinuation of frameworks, as mentioned in the opening text?
- None of the 4 frameworks mentioned in CWIS are capturing the “sanitation economy”. Same for the Principles on Water Governance by OECD.
- There are many other CWIS frameworks that have been developed by World Bank, Eawag, Asian Development Bank.
- Therefore, the main question comes to what is the criteria for including a framework into this list?
- Without this clarity, at least 5 more frameworks could be included, which are currently missing: The Sanitation Cityscape (Scott and Cotton 2020), Guide to Sanitation Resource Recovery (McConville et al., 2020), CWIS Planning Framework (Narayan et al., 2021), WASH benefits accounting framework (CEO Water Mandate, 2024), WASH Impact Investment Indicators (Aqua for all 2023). And when you include the costing tools, there is the PAS WASH costing, IRC Costing and Budgeting tool, Cactus costing tool.
- Table 1: CSDA is not an organisation, right? Please correct this in the table as a lead organisation.
- From the Table 1, seems like iDE, Eawag also have 2 frameworks each.
- From Table 1, the columns are not self-explanatory. Please include a brief line explaining each of these column headings in the caption. For e.g. what does C vs T actually mean in terms of comprehensive vs targeted. And what does ‘Approval’ mean?
- Section 3.2, reads very well as a historical narrative, given the authors’ firsthand experience. It may have some omissions: despite being mired in controversies, the Toilet Board Coalition’s work on the sanitation economy doesn’t seem to feature in this. https://www.toiletboard.org/media/30-Sanitation_Economy_Final.pdf
- Similarly, on CWIS, the call to action was not reportedly signed by 70 organisations, certainly not from the reference you have given here. Perhaps this IWA paper on CWIS is helpful to clarify the narrative: https://iwaponline.com/ebooks/book/934/Discussion-PaperCitywide-Inclusive-Sanitation . Many of the MDBs are also involved in the CWIS approach.
- Section 3.2 overall, could benefit from a short introduction as to why it is useful to know the evolution of WASH enabling environment frameworks. Perhaps it would benefit from relating it more closely and explicitly to the sanitation economy.
- See this Susana publication on an extensive account of the development of urban sanitation frameworks. https://www.susana.org/knowledge-hub/resources?id=4087&nm=2021-11
- The work from SHF can also be included in this section, even though the author affiliations are from SHF. It will not be seen as self-promoting.
- Section 3.4: Include a statement that it may be impossible to know all the places where a tool or framework has been applied or used, and this analysis is only based on publicly available literature.
- Section 3.5: What is meant by impact here? Is it merely uptake and dissemination? Isn’t impact something that is larger than the litmus test proposed here: governments demand that it continues to be used?
- Most of the section 3.5 is all about the dissemination and use of the frameworks. This simply isn’t ‘impact’ per se. Therefore, please reword the sub-title to something more suitable.
- This section 3.5 can also be shortened to have more of a synthesis, rather than a description of the dissemination of these tools. Many of the content in the later section – the current status and future plans of the tools – can be succinctly summarised in a table or as shorter text.
Discussion
- Perhaps it is worth to have a reflection on whether so many frameworks in the WASH sector are useful. Every agency comes up with a new framework from time to time. And reuse of frameworks is quite limited.
- “In 2018, SWA published ‘The SWA Framework’ which incorporates the 736 building blocks and collaborative behaviours [74], embedded within the SWA Mutual 737 Accountability Mechanism [75].” Such sentences are simply a deviation from your main point that the sector is shifting. Otherwise, for it’s inclusion please give a clear link to the argument from this example.
- “Comprehensiveness increases impact.” This is strong statement. But where is the results from the review showing proof of this. Are the more comprehensive frameworks resulting in more impact? This is not seen.
- Similar comment for “Ownership and acceptance lays the foundation for impact.”. For a scientific review, such strong blanket statements need to supported with evidence from the results of the review or from other studies.
- Same for “User friendliness and ease of interpretation increase uptake.”, “If results are not freely available, they will not be used.” “Budgets need to be sustained for frameworks to endure over time.”
- If these statements are meant only to be eye-catching lines, please put them in quotes as subheadings and then present some concrete evidence as to why the above statement is to be taken seriously.
- It may also worth explicitly stating that Climate Frameworks are clearly excluded and therefore considered in this list. Because if you do consider climate frameworks there many more to now start including in the table.
Conclusions
- While the statements made here are fully true, the conclusions do not seem to be built from the evidence and the results of the review. It seems like a standalone piece, that could have well been added to the introduction. A conclusion is meant to answer the objective set out in the paper at the beginning and then a brief outlook.
Author Response
See attached file.
With sincere thanks to the reviewer for the useful comments.

Reviewer 2 Report
Comments and Suggestions for Authors
The article addresses an interesting and important topic for the development of sanitation and the application of WASH (Water, Sanitation, and Hygiene). It aims to review the frameworks used to assess the economics of sanitation in LMICs (low-and middle-income countries), also encompassing broader structures that include WASH. However, the methodology presented is insufficient, being limited to just a few lines and lacking clarity on how the research was conducted. For instance, it mentions that the search was carried out on the internet using specific keywords, but does not specify which databases were consulted—such as Scopus, Web of Science, or Google Scholar. Furthermore, it fails to mention the inclusion and exclusion criteria, and there is no indication that a specific methodology, such as PRISMA, was employed.
In the results section, the article does not state how many documents were found, nor does it clarify the origin or selection process of the 34 frameworks that have been or are currently being used in WASH. Additionally, there is a concern regarding the lack of approval from a research ethics committee. The article presents results that appear to have been obtained through interviews, which makes it essential to disclose whether ethical approval was sought and obtained.
Given the methodological weaknesses, the lack of transparency in the results, and the absence of information regarding ethical approval, I recommend rejecting the article in its current form. It is advised that the authors substantially revise the methodology, clearly describe the study selection procedures, and include appropriate ethics approvals if interviews were indeed conducted. Additionally, the article should present a stronger bibliographic foundation, with references, particularly from more recent scientific articles, that discuss the topic in depth and reflect the current state of the field.
Author Response

(The authors gave the same response as above.)

Round 2
Reviewer 1 Report
Comments and Suggestions for Authors
The paper has been significantly strengthened in this revision version. I am happy to endorse it for publication. Congratulations to the authors!